# Deep Learning Approach to Classify Cutaneous Melanoma in a Whole Slide Image

**DOI:** 10.3390/cancers15061907

**Published:** 2023-03-22

**Authors:** Meng Li, Makoto Abe, Shigeo Nakano, Masayuki Tsuneki

**Affiliations:** 1Medmain Research, Medmain Inc., Fukuoka 810-0042, Japan; 2Department of Pathology, Tochigi Cancer Center, 4-9-13 Yohnan, Utsunomiya 320-0834, Japan; 3Department of Surgical Pathology, Tokyo Shinagawa Hospital, 6-3-22 Higashi-Ooi, Shinagawa, Tokyo 140-8522, Japan

**Keywords:** melanoma, deep learning, cancer screening, whole slide image

## Abstract

**Simple Summary:**

In this paper, we investigate the application of deep learning for classifying whole-slide images of cutaneous histopathological specimens into melanoma and non-melanoma. To do so, we used a total of 66 images (33 melanomas and 33 non-melanomas) to train models and evaluated them on 90 whole-slide images (40 melanomas and 50 non-melanomas). The best model achieved ROC–AUC at 0.821 for the whole-slide image level and 0.936 for the tile level.

**Abstract:**

Although the histopathological diagnosis of cutaneous melanocytic lesions is fairly accurate and reliable among experienced surgical pathologists, it is not perfect in every case (especially melanoma). Microscopic examination–clinicopathological correlation is the gold standard for the definitive diagnosis of melanoma. Pathologists may encounter diagnostic controversies when melanoma closely mimics Spitz’s nevus or blue nevus, exhibits amelanotic histopathology, or is in situ. It would be beneficial if diagnosing cutaneous melanocytic lesions can be automated by using deep learning, particularly when assisting surgical pathologists with their workloads. In this preliminary study, we investigated the application of deep learning for classifying cutaneous melanoma in whole-slide images (WSIs). We trained models via weakly supervised learning using a dataset of 66 WSIs (33 melanomas and 33 non-melanomas). We evaluated the models on a test set of 90 WSIs (40 melanomas and 50 non-melanomas), achieving ROC–AUC at 0.821 for the WSI level and 0.936 for the tile level by the best model.

## 1. Introduction

The Global Cancer Statistics 2020 report noted that there were 324,635 new cases of cutaneous melanoma (1.7% of all cancer sites) and 57,043 new deaths (0.6% of all cancer sites) worldwide [1]. When comparing 2020 to 2010, there were 68,130 newly diagnosed melanoma cases [2]. Melanoma is one of the most common cancers with increasing incidence rates in the United States, but relatively rare incidences in Japan [3,4]. Approximately 30% of cutaneous melanomas develop in conjunction with a nevus (nevus-associated melanoma), supporting the concept of dysplastic nevi and common-acquired nevi as precursors to cutaneous melanoma [5,6]. In addition, lesions diagnosed as simple lentigo or solar lentigo are potentially important precursors of melanoma [6]. The remaining 70% generally develop de novo from clinically normal skin [5,7]. Pathological examination is considered the gold standard for diagnosing cutaneous melanoma [2], with a typical case being identified by certain distinctive characteristics, such as invasive activity with neighboring cells, marked melanin pigmentation, and marked cellular and nuclear atypia with mitotic figures [8,9].

There are three major issues in the histopathological diagnosis of melanoma. First, melanoma is notorious for its great microscopic variability [10,11]. For example, melanoma cells can resemble epithelial cells, or have shapes that are spindled or extremely bizarre (monster cells) [10,12]; they can range from small to large with multiple nuclei [13]. The cytoplasms of melanoma cells can be eosinophilic, basophilic, foamy, rhabdoid, oncocytic, or clear (balloon cells) [14,15,16,17,18,19]. Melanin can be abundant, insufficient, or absent (amelanotic melanoma) [20].

Second, melanoma and benign nevus share many characteristics, leading to some confusion during histopathological diagnosis [21]. For example, blue nevus, combined nevus, deep penetrating nevus, and atypical Spitz nevus exhibit unusual maturation patterns [21]. Atypical Spitz nevus can also exhibit cellular atypia and poor circumscription [21]. These are all benign lesions and, therefore, do not require aggressive treatment. Pathologists can avoid misdiagnosing melanoma by being familiar with these lesions.

Third, one of the most controversial aspects in the histopathological diagnosis is melanoma in situ—melanoma lesions that are limited to the epidermis [2]. Typically, melanoma grows horizontally within the epidermis and then penetrates the dermis [2]. Detecting melanoma at an early stage of its evolution (melanoma in situ) is important for saving lives [2].

Therefore, histopathological diagnosis is critical for melanoma, especially in differentiating between melanoma, melanoma in situ, and nevus, as this can have a significant impact on treatment.

Performing primary diagnosis using whole-slide images (WSIs) is considered to be similar to microscopy [22,23]. Using WSIs can help reduce the pathologist’s working time by providing convenient access to high-quality pathological images via cloud-based software; this saves on resources and costs by eliminating sliding-glass shipping expenses [23]. Deep learning applications on WSIs have shown great promise in the past few years for the creation of new tools in assisting pathologists [24]. Previous studies have looked into melanoma classification and segmentation; for instance, [25] used 39 cases to classify cases into melanocytic nevi, Spitz nevi, and invasive melanoma; while [26] trained a model for segmentation of the nuclei in melanoma cases.

In this study, we trained convolutional neural networks (CNNs) by using a training dataset consisting of 33 cutaneous melanoma and 33 cutaneous non-melanoma lesion WSIs. We evaluated the models on a test set of 90 WSIs (40 melanomas and 50 non-melanomas), achieving ROC–AUCs for the WSI evaluation in the range of 0.700–0.825 and the tile-level evaluation in the range of 0.887–0.936.

## 2. Materials and Methods

### 2.1. Clinical Cases and Histopathological Records

In this retrospective study, we initially retrieved 751 histopathological H&E (hematoxylin and eosin)-stained sliding-glass cutaneous specimens from the surgical pathology files of Kamachi Group Hospitals (Shintakeo, Shinmizumaki, Wajiro, and Tokyo Shinagawa hospitals) (Fukuoka and Tokyo, Japan). The glass slides were digitized into WSIs at a ×20 magnification using a Leica Aperio AT2 Digital Whole Slide Scanner (Leica Biosystems, Tokyo, Japan).

After a histopathological review of all specimens by three expert surgical pathologists, 97 were determined to have melanoma diagnoses. Out of these 97 cases, 19 were excluded (14 melanoma in situ and 5 invasive melanoma cases) due to diagnostic inconsistency. This resulted in a final number of 78 melanoma cases. In order to have a balanced set for training and testing, we selected a roughly equal amount of 88 non-melanoma cases. Thus, in total, we had 166 WSIs (78 melanoma and 88 non-melanoma). Table 1 and Table 2 provide a breakdown of the composition of the cases, and Table 3 provides a breakdown of cases into training, validation, and test sets.

### 2.2. Annotation

Pathologists who performed routine histopathological diagnoses in general hospitals in Japan manually annotated 78 melanoma WSIs from training (33 WSIs), validation (5 WSIs), and test (40 WSIs) sets (Table 3). Coarse polygonal annotations were obtained by manual hand drawings using an in-house online tool developed by customizing the open-source OpenSeadragon tool at https://openseadragon.github.io/ (accessed on 21 February 2021), which is a web-based viewer for zoomable images. Overall, the pathologists manually annotated melanoma cell-infiltrating area on WSIs (Figure 1). We set an annotation label as melanoma (Figure 1). Due to the tile-level evaluation, we annotated the melanoma test set (Table 3, Figure 1). The non-melanoma subsets of the training, validation, and test sets (Table 3) were not annotated and the entire tissue areas within the WSIs were used. The average annotation time per WSI was about 15 min. The annotations performed by the pathologists were modified (if necessary) and verified by a senior pathologist.

### 2.3. Deep Learning Models

We used a modified version of the EfficientNetB1 (CNN) architecture. We performed training using weakly supervised (WS) learning in a manner similar to the one described in [27]. During slide-tiling, a given WSI at a magnification of ×20 was sliced into overlapping tiles through a sliding window. We used 4 sets of tiling sizes: 224 × 224 px with a stride of 224 × 224 px, 512 × 512 px with a stride of 256 × 256 px, 768 × 768 px with a stride of 512 × 512 px, and 1024 × 1024 px with a stride of 512 × 512 px. This resulted in four deep learning models.

We first detected tissue regions in a WSI by thresholding the grayscale version of the WSI through Otsu’s method to exclude the white background. Annotations were used afterward when available to further reduce the valid tissue regions to the annotated tissue regions. Then, we divided the WSI into a grid and sampled tiles using the grid cell locations from the valid tissue regions. When sampling, we performed image augmentation of the extracted tiles using a variety of image transformations related to the color and image quality. This allows the model to focus more on the content. We did not apply ICC profile transformation to WSIs because they were from the same scanner and shared the same color space. It might be necessary to apply the ICC profile and gamma transformation to WSIs from different scanners to eliminate the impact of color space differences.

During training, we carried out a balanced random sampling of tiles from all WSIs in each epoch to ensure that there was no class imbalance. For melanoma WSIs, we randomly sampled from the annotated tissue regions; for non-melanoma, we randomly sampled from all of the valid tissue regions. After a few epochs, we switched between the training and inference stages and updated the mechanism of generating the subset, which was used in the training stage. During inference, we applied the CNN to all of the tiles extracted from the tissue regions in WSIs. We then selected the top *k* tiles that were most likely to be false positives when the true label of the WSI was non-melanoma.

During the gradient descent procedure, we used the Adam (adaptive moment estimation) optimizer with the binary cross-entropy loss function, with the parameters configured as follows: beta1 = 0.9, beta2 = 0.999, batch size = (32 for 224 × 224 px, 16 for 512 × 512 px, 8 for 768 × 768 px and 1024 × 1024 px), and the initial learning rate = 0.001 when fine tuning. We used early stopping by monitoring the performance improvement of the model on a validation set, and the training was automatically stopped when the validation loss no longer improved for 10 epochs.

During the evaluation stage, we obtained a WSI level prediction for each WSI by simply taking the maximum probability among all of its tiles.

For a bottom-to-up comparison, we evaluated the metrics of all tile-level predictions. The metrics included ROC–AUC, log-loss, accuracy, sensitivity (true positive rate TPR), specificity (true negative rate TNR), precision (positive prediction value PPV), etc. We calculated the 95% confidence intervals (CIs) of the metrics by performing bootstrap resampling, with a reduced resample count compared to the one used in the WSI-level evaluation because of the increased dataset size and computation costs. We finally chose the model that had the highest accuracy among the tile-level evaluation results and equivalent scores as other models at the WSI level.

### 2.4. Software and Statistical Analysis

We used the open-source TensorFlow library [28] to implement the models. We calculated the AUCs using the scikit-learn package [29] and plotted using matplotlib [30]. We computed the 95% CIs of the AUCs using the bootstrap method [31] with 1000 iterations.

## 3. Results

### 3.1. High ROC–AUC Performance of Melanoma WSI and Tile-Level Evaluation

We trained four deep learning models (×20, 224 × 224 px; ×20, 512 × 512 px; ×20, 768 × 768 px; ×20, 1024 × 1024 px), the primary difference between them being the tile size. We evaluated these four models on the test set (Table 3) at the WSI and tile levels by computing the ROC–AUC, log-loss, accuracy, sensitivity, specificity, and confusion matrix. We summarized the results in Table 4 and Table 5 and Figure 2. Overall, the model (×20, 512 × 512 px) achieved the highest ROC–AUC (0.936) at the tile level and equivalent scores as the models (×20, 768 × 768 px and ×20, 1024 × 1024 px) at the WSI level.

### 3.2. True Positive Prediction

The model (×20, 512 × 512 px) satisfactorily predicted melanoma cells with melanin in WSIs (Figure 3). According to the pathologists’ markings (black triangles), melanoma cells can be observed in the central area of WSI (in between black triangles) (Figure 3A,C,E). Consistent with the distribution of melanoma cells (Figure 3A,C,E), Figure 3B showed true positive predictions of melanoma cells (Figure 3D,F). Compared to the probabilities of the central area of melanoma (Figure 3C), which enriches melanoma cells (Figure 3D), the marginal area (Figure 3E) shows lower probabilities (Figure 3F), which reflects the number of melanoma cells. At the tile level, melanoma cells with melanin (Figure 4A) were predicted satisfactorily by the model (×20, 512 × 512 px) (Figure 4B).

### 3.3. True Negative Prediction

The model (×20, 512 × 512 px) satisfactorily predicts non-melanoma cells, especially melanocytic nevus, which includes melanin (Figure 5). The heatmap images show the true negative predictions of melanocytic nevus (Figure 5B,D). Regarding the tile-level evaluation of melanocytic nevus (Figure 4C), the heatmap image shows true negative predictions of melanin-positive and -negative melanocytic nevus cells (Figure 4D).

### 3.4. False Positive Prediction

Out of nine melanoma false positives predicted by the model in WSIs that were (×20, 512 × 512 px) (Table 5), six out of nine WSIs (66.7%) were histopathologically diagnosed as Spitz’s nevus (Figure 6A) and three out of nine WSIs (33.3%) were blue nevi (Figure 6E), which are histopathological mimickers of melanoma. The heatmap images show false positive predictions of melanin-positive Spitz’s nevus (Figure 6B–D) and blue nevus (Figure 6F–H) cells.

### 3.5. False Negative Prediction

At the WSI-level evaluation by the model (×20, 512 × 512 px) (Table 5), all eleven WSIs that were false-negatively predicted were histopathologically diagnosed as melanoma in situ (Figure 7A,C). The heatmap images show low melanoma-predicted tiles in the area of melanoma in situ (Figure 7B–D). However, t the WSI level, melanoma in situ WSI was false-positively predicted. At the tile level, amelanotic variants of melanoma cells without melanin (Figure 8A,C,E,G) were false-negatively predicted (Figure 8B,D,F,H).

## 4. Discussion

In this pilot study, we trained deep learning models for the classification of cutaneous melanoma in WSIs. We computed metric scores at the WSI and tile levels. The best model (×20, 512 × 512 px) achieved ROC–AUC of 0.821 (CI: 0.712–0.890) at the WSI level, 0.936 (CI: 0.935–0.938) at the tile level, and low log-loss values of 0.532 (CI: 0.472–0.618) at the WSI level and 0.151 (CI: 0.149–0.153) at the tile level (Table 4). The heatmap images reveal that the best model (×20, 512 × 512 px) true-positively predicted cutaneous melanoma cells with melanin (Figure 3) and true-negatively predicted non-melanoma (melanocytic nevus) with melanin (Figure 5). The model (×20, 512 × 512 px) could differentiate between melanoma and non-melanoma (nevi) with melanin satisfactorily (Figure 4).

This study was challenging because of the limited number of cases that could be collected (Table 3). To train a reliable model and evaluate it accurately, we would need a dataset of cutaneous WSIs, which are diagnosed as melanoma or non-melanoma; however, this is challenging because there are many controversial cases in cutaneous lesions. In addition, there are three other major limitations in this study: First, Spitz’s nevus and blue nevus were false-positively predicted as melanoma by the model (×20, 512 × 512 px). Out of 9 false positive WSIs (Table 5), 6 WSIs (66.7%) were Spitz’s nevi and 3 WSIs (33.3%) were blue nevi. Spitz’s nevus typically affects children and young adults, especially on the face and lower extremities, but it may occur anywhere and at any age [32].

Histopathologically, Spitz’s nevus has many similarities to melanoma morphology [21,32], and we see this exhibited with the false positive cases: melanocytes dispersed in single-cell patterns, circumscribed lesions, and nests present at edges rather than single cells. Moreover, there are Spitz’s melanomas which are expected to show Spitz-like features that could be classified as malignant melanomas [32,33,34]. The diagnosis of melanoma may be based on a clinical follow-up and/or the opinion of an experienced pathologist [32]. In this study, Spitz’s nevus was not included in the datasets (both training and validation sets). Spitz’s nevus is one of the diagnostic pitfalls in cutaneous melanocytic lesions for pathologists [34]. The blue nevus is usually small and located in the head and neck or upper extremity [35]. Histopathologically, the blue nevus also has many similarities to melanoma morphology; it is typically characterized by an indistinct deep dermal spread of long and dendritic dermal melanocytes with abundant melanin [21,35]. Metastatic melanoma from the skin (or eye) is the most important differential diagnosis of blue nevus [36,37]. Based on these histopathological features, it is understandable that Spitz’s and blue nevi were false-positively predicted as melanoma (Figure 6).

Second, at the tile level, amelanotic variants of melanoma cells without melanin were false-negatively predicted as melanoma by the model (×20, 512 × 512 px) (Figure 8). When a lesion is amelanotic, melanoma can be confused with a number of non-melanocytic tumors [10,11]. For example, based solely on hematoxylin and eosin (H & E)-stained specimens, benign tumors such as foam and giant cell-poor xanthogranuloma or fibroma can be misdiagnosed as melanoma [21]. In routine histopathological diagnoses, immunohistochemical studies allow for definitive differentiation between them. Therefore, it is understandable that it was difficult to precisely predict amelanotic variants of melanoma (Figure 8) at the tile level.

Third, it was hard to predict melanoma in situ at the WSI level (Figure 7). All eleven false negative WSIs were melanoma in situ. However, in routine histopathological diagnoses, melanoma in situ is one of the most controversial aspects in the pathology of melanocytic lesions [21,38,39,40].

The major three limitations mentioned previously are due to a lack of training datasets that cover controversial nevi (e.g., Spitz’s nevus and blue nevus), amelanotic melanomas, and melanoma in situ. Thus, for future work, we will attempt to collect more melanocytic lesions from a wide variety of hospitals and clinical laboratories and perform active and/or iterative learning on the existing model (×20, 512 × 512 px).

## 5. Conclusions

In this study, we trained models to classify cutaneous melanoma WSIs, and we evaluated them at the WSI and tile levels, achieving ROC–AUCs for the WSI level at 0.821 (CI: 0.712–0.890) and the tile level at 0.936 (CI: 0.935–0.938) by the best model (×20, 512 × 512 px). At the WSI and tile levels, the model (×20, 512 × 512 px) satisfactorily predicted melanoma with melanin and true-negatively predicted melanin-positive nevus by the heatmap images. At the WSI level, the model (×20, 512 × 512 px) false-positively predicted Spitz’s nevus and blue nevus and false-negatively predicted melanoma in situ. At the tile level, the model (×20, 512 × 512 px) false-negatively predicted amelanotic melanoma cells that lacked melanin. To overcome these false positives and false negatives, the number of images in the training set should be increased.

## Figures and Tables

**Figure 1 cancers-15-01907-f001:**
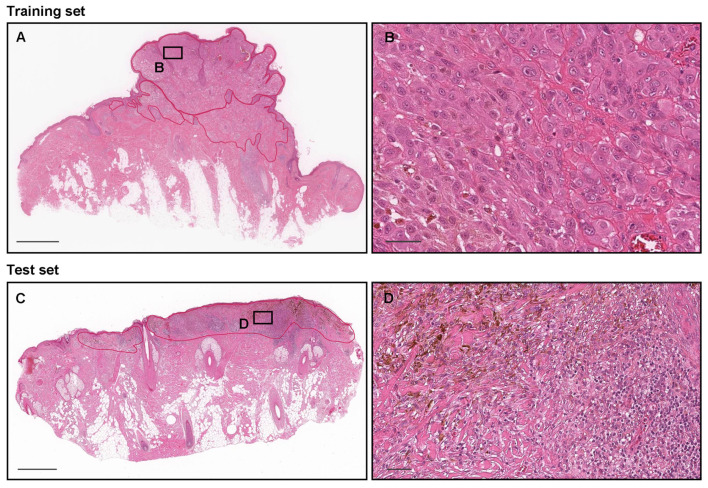
A manual drawing of an annotation for melanoma labels on whole-slide images (WSIs). In both training (**A**) and test (**C**) sets, the melanoma cell-infiltrating areas (**B**,**D**) were manually annotated (red lines) as melanoma labels. The test set of melanoma WSIs was annotated due to the tile-based evaluation. Scale bars: 1 mm (**A**,**C**) and 50 μm (**B**,**D**).

**Figure 2 cancers-15-01907-f002:**
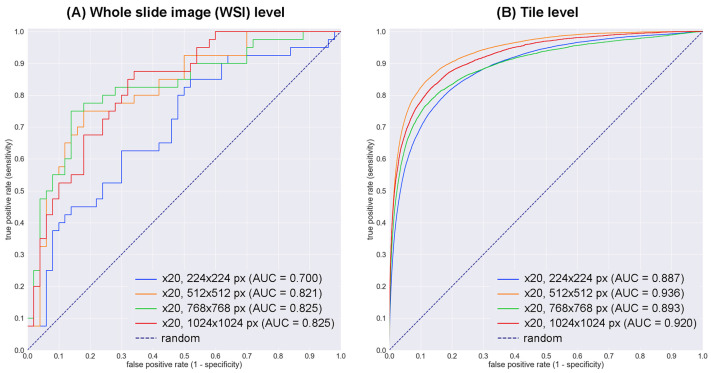
ROC curves for the test set using four trained deep learning models at the whole-slide image (WSI) level and tile-level evaluations. At the WSI level evaluation (**A**), the models ([×20, 512 × 512 px], [×20, 768 × 768 px], and [×20, 1024 × 1024 px]) showed near-equivalent area under the curve (AUC) scores, which reached 0.821–0.825. With the tile-level evaluation (**B**), the model (×20, 512 × 512 px) exhibited the highest AUC score (0.936).

**Figure 3 cancers-15-01907-f003:**
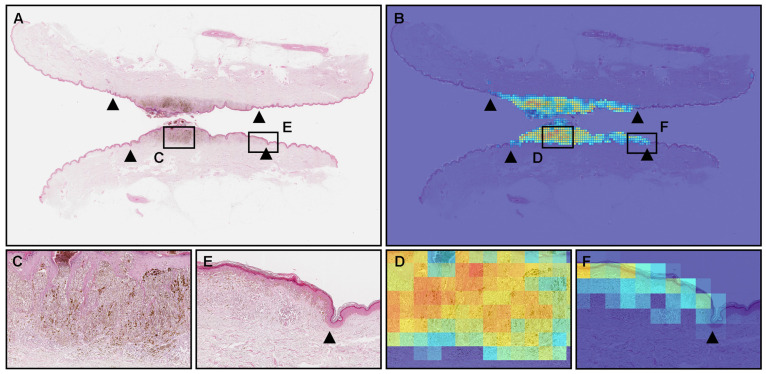
An example of the true positive predictions of melanoma; consistent with the distribution of the melanoma lesion as indicated by the pathologist (black triangles) (**A**), the heatmap image (**B**) shows the true positive predictions of the melanoma area. The model (×20, 512 × 512 px) predicted satisfactorily melanoma cells, including melanin (**C**,**D**). In the marginal area (**E**,**F**)), the model (×20, 512 × 512 px) could detect a small number of melanoma cells. Red indicates a high probability and blue indicates a low probability.

**Figure 4 cancers-15-01907-f004:**
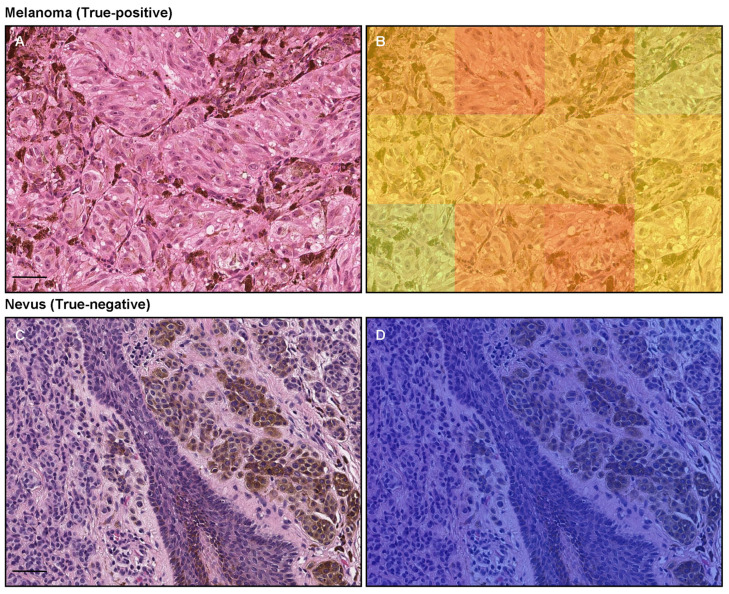
An example of true positive and true negative predictions of melanoma using the ×20, 512 × 512 px model. (**A**) was a melanoma area with melanin, which was true-positively predicted as melanoma by the heatmap image (**B**). (**C**) was a melanocytic nevus area with and without melanin, which was true-negatively predicted as melanoma by the heatmap image (**D**). Red indicates a high probability and blue indicates a low probability. Scale bars: 50 μm.

**Figure 5 cancers-15-01907-f005:**
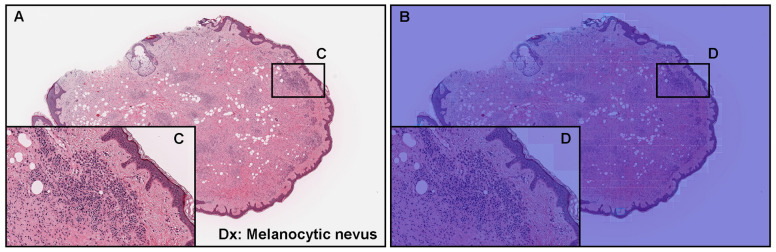
An example of true negative predictions of melanoma using the ×20, 512 × 512 px model. According to the histopathological diagnostic (Dx) report, (**A**) was diagnosed as melanocytic nevus according to the representative morphology (**C**). The heatmap images (**B**,**D**) show true negative predictions of melanoma cells. Red indicates a high probability and blue indicates a low probability.

**Figure 6 cancers-15-01907-f006:**
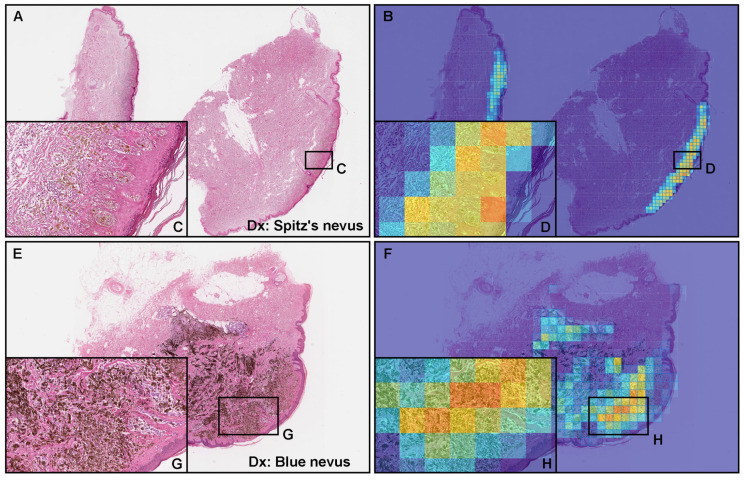
Two examples of false positive predictions of melanoma using the ×20, 512 × 512 px model. According to the histopathological diagnostic (Dx) reports, (**A**) was diagnosed as a Spitz nevus and (**E**) a blue nevus. Both Spitz’s nevus cells (**C**) and blue nevus cells (**G**) are morphological mimics of melanoma cells. The heatmap images (**B**,**F**) show false positive predictions and (**D**,**H**) correspond, respectively, to (**C**,**G**). Red indicates a high probability and blue indicates a low probability.

**Figure 7 cancers-15-01907-f007:**
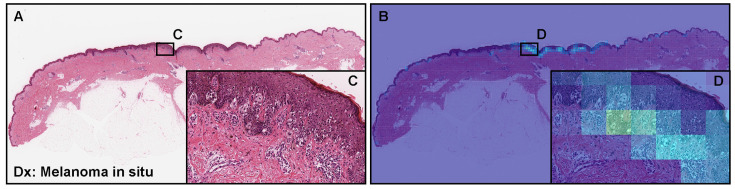
An example of melanoma false negative prediction using the ×20, 512 × 512 px model. According to the histopathological diagnostic (Dx) reports, (**A**) was diagnosed as melanoma in situ, which is also called stage 0 melanoma. Melanoma cells are found in the epidermis (**C**). The heatmap image (**B**,**D**) shows very low probabilities of melanoma but false negative predictions at the WSI level. Red indicates a high probability and blue indicates a low probability.

**Figure 8 cancers-15-01907-f008:**
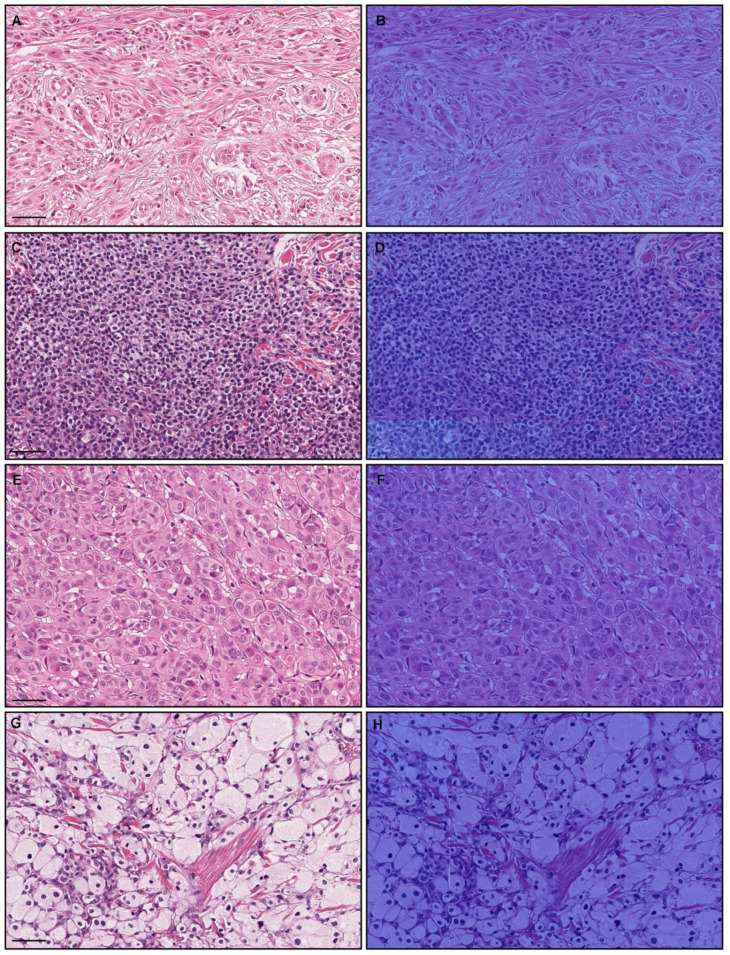
An example of melanoma false negative using the ×20, 512 × 512 px model. Histopathologically amelanotic variants of melanoma cells without melanin (**A**,**C**,**E**,**G**) were false-negatively predicted as melanoma (**B**,**D**,**F**,**H**) at the tile level. Red indicates a high probability and blue indicates a low probability. Scale bars: 50 μm.

**Table 1 cancers-15-01907-t001:** The composition of the 78 melanoma subtype cases.

Subtype	Site	WSI	Subtype	Site	WSI
Melanoma in-situ	Head and neck	4	NOS	Head and neck	3
	Upper extremity	4		Trunk	3
	Lower extremity	6		Upper extremity	2
Nodular	Head and neck	3		Lower extremity	3
	Trunk	5	Amelanotic	Head and neck	4
	Upper extremity	7		Trunk	3
	Lower extremity	8		Upper extremity	5
Lentigo maligna	Head and neck	5		Lower extremity	4
	Upper extremity	1			
Superficial spreading	Head and neck	2			
	Trunk	3			
	Upper extremity	2			
	Lower extremity	1			

**Table 2 cancers-15-01907-t002:** The composition of the 88 non-melanoma subtype cases.

Subtype	Site	WSI	Subtype	Site	WSI
Compound nevus	Head and neck	6	Spitz’s nevus	Head and neck	5
	Trunk	4		Lower extremity	1
	Upper extremity	6	Congenital nevus	Head and neck	1
	Lower extremity	5		Upper extremity	1
Junctional nevus	Head and neck	5	Normal skin	Head and neck	3
	Trunk	3		Trunk	3
	Upper extremity	5		Upper extremity	4
	Lower extremity	4		Lower extremity	3
Intradermal nevus	Head and neck	3	Non-melanocytic benign	Head and neck	3
	Trunk	1		Trunk	4
	Upper extremity	2		Upper extremity	3
	Lower extremity	3		Lower extremity	4
Blue nevus	Head and neck	2			
	Trunk	1			
	Upper extremity	1			
	Lower extremity	2			

**Table 3 cancers-15-01907-t003:** Datasets.

	Melanoma	Non-Melanoma
Training	33	33
Validation	5	5
Test	40	50
Total	78	88

**Table 4 cancers-15-01907-t004:** Results of the test set of whole-slide image (WSI) level and tile level using multiple metrics.

	Evaluation	
	WSI Level	Tile Level
*×20, 224 × 224 px*		
ROC–AUC	0.700 [0.587–0.808]	0.887 [0.885–0.890]
Log-loss	0.666 [0.642–0.686]	0.328 [0.327–0.329]
Accuracy	0.644 [0.544–0.744]	0.833 [0.831–0.837]
Sensitivity	0.850 [0.705–0.933]	0.788 [0.783–0.793]
Specificity	0.480 [0.366–0.647]	0.835 [0.834–0.837]
*×20, 512 × 512 px*		
ROC–AUC	0.821 [0.712–0.890]	0.936 [0.935–0.938]
Log-loss	0.532 [0.472–0.618]	0.151 [0.149–0.153]
Accuracy	0.778 [0.667–0.844]	0.881 [0.880–0.882]
Sensitivity	0.725 [0.538–0.844]	0.844 [0.839–0.849]
Specificity	0.820 [0.691–0.902]	0.883 [0.882–0.884]
*×20, 768 × 768 px*		
ROC–AUC	0.825 [0.763–0.930]	0.893 [0.888–0.898]
Log-loss	0.568 [0.435–0.651]	0.171 [0.165–0.174]
Accuracy	0.811 [0.767–0.911]	0.860 [0.858–0.862]
Sensitivity	0.750 [0.667–0.920]	0.786 [0.777–0.798]
Specificity	0.860 [0.786–0.964]	0.865 [0.863–0.867]
*×20, 1024 × 1024 px*		
ROC–AUC	0.825 [0.752–0.916]	0.920 [0.918–0.924]
Log-loss	0.577 [0.383–0.704]	0.186 [0.180–0.192]
Accuracy	0.756 [0.667–0.844]	0.863 [0.860–0.865]
Sensitivity	0.875 [0.755–0.961]	0.823 [0.817–0.835]
Specificity	0.660 [0.523–0.800]	0.865 [0.863–0.868]

**Table 5 cancers-15-01907-t005:** Confusion matrix on both the WSI level and tile level. A cut-off threshold of 50% was used to convert the probabilities into classifications.

		WSI Level		Tile Level	
		Predicted Label		Predicted Label	
		Melanoma	Non-Melanoma	Melanoma	Non-Melanoma
*×20, 224 × 224 px*					
True label	Melanoma	34	6	22,241	5990
	Non-melanoma	26	24	70,527	358,037
*×20, 512 × 512 px*					
True label	Melanoma	29	11	18,706	3471
	Non-melanoma	9	41	39,055	294,926
*×20, 768 × 768 px*					
True label	Melanoma	30	10	1287	653
	Non-melanoma	7	43	46,007	38,460
*×20, 1024 × 1024 px*					
True label	Melanoma	35	5	5621	1206
	Non-melanoma	17	33	12,996	83,448

## Data Availability

The datasets generated during and/or analyzed during the current study are not publicly available due to the specific institutional requirements governing privacy protection but are available from the corresponding author upon reasonable request. The datasets that support the findings of this study are available from Kamachi Group Hospitals (Fukuoka, Japan), but restrictions apply to the availability of these data, which were used under a data-use agreement that was made according to the Ethical Guidelines for Medical and Health Research Involving Human Subjects as set by the Japanese Ministry of Health, Labour, and Welfare (Tokyo, Japan) and, thus, are not publicly available. However, the data are available from the authors upon reasonable request for private viewing and with permission from the corresponding medical institutions within the terms of the data-use agreement and if compliant with the ethical and legal requirements as stipulated by the Japanese Ministry of Health, Labour, and Welfare.

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
