# Peer review of "Deep Learning Approach to Classify Cutaneous Melanoma in a Whole Slide Image"

_cancers, 2023, doi:10.3390/cancers15061907_

Round 1

Reviewer 1 Report

Thank you for this interesting manuscript on applying deep learning approaches to histopathologic diagnosis of melanoma. The authors leverage a small cohort of melanoma and non-melanoma patients from Japan to train and validate an ensemble CNN architecture in differentiating between malignant and non-malignant tissue using tile level and whole slide image (WSI) level approaches. They find that tile level approaches outperformed WSI approaches, but that both had reasonably high performance ranging from AUC of 0.82 to 0.94. The overall approach is sound and follows previous  investigations of deep learning approaches to histologic diagnosis in melanoma and non-melanoma cancer settings. However, several items need to be addressed prior to the publication of this manuscript.

Major:

1. The authors provide little to no detail on how the cases and controls were selected. Were these selected as a convenience sample from the included institutions? What quality control, if any, was performed in the inclusion of the final slides for analysis? Who made the determination of which cases should be excluded, if any?

2. There is insufficient detail on the composition of the control population. Were all of these benign melanocytic lesions (e.g. benign nevi)? Did some include dysplastic nevi (and what type - mild, moderate, severe)? Did some include normal skin or non-melanocytic benign lesions? The authors should provide a detailed baseline characteristics table with demographic, anatomic, and lesion-specific information on the control population.

3. Please also provide similar information on the composition of the included melanomas. The authors allude that some of these with MIS, while others were invasive. Please provide a detailed breakdown of this split as well as information on melanoma subtype (e.g. nodular, superficial spreading, lentigo maligna, melanoma NOS, amelanotic melanoma, etc) for the invasive lesions. It would also be helpful to include the anatomic location from where the biopsies were performed (trunk, extremity, face, acral locations, etc) for both cases and controls and how these influenced model prediction.

4. What minimum probability threshold was used to decide if a case was melanoma in WSI-based models? How does the overall model performance change if this threshold was to vary?

5. On page 5 of the manuscript, the following statements are conflicting: "Out of 9 false positive WSIs (Table 3), 6 WSIs (66.7%) were Spitz’s nevus and 3 WSIs (33.3%) were blue nevus" and "In this study, the Spitz’s nevus was not included in the dataset (both training and test sets)." Were spitz nevi included or not included in the study?

6. What, if any, clinical information was included in the prediction? How would the models improve if such information were to be included (e.g age, sex, anatomic location of lesion, etc).

7.  Please include a discussion on how these model results compare with previous reports of deep learning tools in melanoma histopathology evaluation.

8. What was the rate of diagnostic concordance between independent pathologists reviewing and annotating cases and controls? Were they blinded to the case status upon review and annotation? This is a significant area of concern in the literature as the overall diagnostic concordance of benign vs malignant melanocytic lesions is fairly poor among pathologists and even temporally within the same pathologist (when asked to review the same case at two different times). Given that pathology review is considered a gold standard for evaluating model performance, it is necessary to comment on this issue and how it affects the current analysis. What was the overall reliability in the diagnoses?

Author Response

Reviewer 1:

Thank you for this interesting manuscript on applying deep learning approaches to histopathologic diagnosis of melanoma. The authors leverage a small cohort of melanoma and non-melanoma patients from Japan to train and validate an ensemble CNN architecture in differentiating between malignant and non-malignant tissue using tile level and whole slide image (WSI) level approaches. They find that tile level approaches outperformed WSI approaches, but that both had reasonably high performance ranging from AUC of 0.82 to 0.94. The overall approach is sound and follows previous  investigations of deep learning approaches to histologic diagnosis in melanoma and non-melanoma cancer settings. However, several items need to be addressed prior to the publication of this manuscript.

Major:

  1. The authors provide little to no detail on how the cases and controls were selected. Were these selected as a convenience sample from the included institutions? What quality control, if any, was performed in the inclusion of the final slides for analysis? Who made the determination of which cases should be excluded, if any?

Response: We have added a clarification in the “Clinical Cases and histopathological Records” section.

  1. There is insufficient detail on the composition of the control population. Were all of these benign melanocytic lesions (e.g. benign nevi)? Did some include dysplastic nevi (and what type - mild, moderate, severe)? Did some include normal skin or non-melanocytic benign lesions? The authors should provide a detailed baseline characteristics table with demographic, anatomic, and lesion-specific information on the control population.
  2. Please also provide similar information on the composition of the included melanomas. The authors allude that some of these with MIS, while others were invasive. Please provide a detailed breakdown of this split as well as information on melanoma subtype (e.g. nodular, superficial spreading, lentigo maligna, melanoma NOS, amelanotic melanoma, etc) for the invasive lesions. It would also be helpful to include the anatomic location from where the biopsies were performed (trunk, extremity, face, acral locations, etc) for both cases and controls and how these influenced model prediction.

Response: We have added two additional tables that provide a breakdown of the composition of melanoma and non-melanoma cases.

  1. What minimum probability threshold was used to decide if a case was melanoma in WSI-based models? How does the overall model performance change if this threshold was to vary?

Response: We used the standard 50% threshold for computing the  accuracy, specificity, specificity, and log loss. The ROC curves plot how the sensitivity and specificity vary with the threshold going from 0% to 100%.

  1. On page 5 of the manuscript, the following statements are conflicting: "Out of 9 false positive WSIs (Table 3), 6 WSIs (66.7%) were Spitz’s nevus and 3 WSIs (33.3%) were blue nevus" and "In this study, the Spitz’s nevus was not included in the dataset (both training and test sets)." Were spitz nevi included or not included in the study?

Response: Sorry, this was a mistake, we meant  “both training and validation sets”. We have corrected this.

  1. What, if any, clinical information was included in the prediction? How would the models improve if such information were to be included (e.g age, sex, anatomic location of lesion, etc).

Response: No clinical information was included as we wanted only to attempt prediction from WSIs.

  1. Please include a discussion on how these model results compare with previous reports of deep learning tools in melanoma histopathology evaluation.

Response: We have added in the introduction some references to some recent reports of melanoma classification and segmentation.

  1. What was the rate of diagnostic concordance between independent pathologists reviewing and annotating cases and controls? Were they blinded to the case status upon review and annotation? This is a significant area of concern in the literature as the overall diagnostic concordance of benign vs malignant melanocytic lesions is fairly poor among pathologists and even temporally within the same pathologist (when asked to review the same case at two different times). Given that pathology review is considered a gold standard for evaluating model performance, it is necessary to comment on this issue and how it affects the current analysis. What was the overall reliability in the diagnoses?

Response: During the reviewing process, the pathologists were blinded to the case status.

We have excluded 19 melanoma WSIs (14 melanoma in-situ and 5 invasive melanoma) due to diagnostic inconsistency. As for the annotation, we have selected the annotation area where the annotated pathologist and senior pathologist made agreement. 

Reviewer 2 Report

The methodology followed to detect cutaneous melanoma from whole slide images is an interesting research. Dataset to work on, is represented as one of the challenges in the paper. Yes I do feel it to be the  most important challenge.

1. with such small dataset to deep learning model, the results obtained cannot be trusted. The confusion matrix given in table 3 is not stable and seems.

2. since the dataset is small and to check the stability of the results obtained, the authors can do five or more cross fold validation.

3.Deep learning model/architecture or some kind of block /flow diagram to represent the flow of the implementation to be included.

Author Response

Reviewer 2:

The methodology followed to detect cutaneous melanoma from whole slide images is an interesting research. Dataset to work on, is represented as one of the challenges in the paper. Yes I do feel it to be the  most important challenge.

  1. with such small dataset to deep learning model, the results obtained cannot be trusted. The confusion matrix given in table 3 is not stable and seems.

Response: Melanoma cases are very hard to obtain in large quantities. In addition, whole slide images are extremely large. You can note in Table 3 the tile level evaluation where there were more than 400K tiles – which is a large number.

  1. since the dataset is small and to check the stability of the results obtained, the authors can do five or more cross fold validation.

Response: While the number of WSIs is small, a given WSI is extremely large. Each WSI is about 40,000x60,000 pixels on average. As these need to be broken down into tiles, they generate from a few hundred thousand to a few million tiles – this is a large number of images. It is also why we report the results on  both the slide level and tile level. In addition, we performed bootstrapping [1] instead of cross validation to obtain confidence interval estimates. Bootstrapping is an alternative method to cross validation, and is also commonly used. References [2,3] are two publications in high impact journals (Impact factor > 40) that deal with a similar application and do not perform cross validation, and instead use the bootstrapping method. 

[1] Efron, B.; Tibshirani, R.J. An introduction to the bootstrap; CRC press, 1994

[2] Bulten, Wouter, et al. "Automated deep-learning system for Gleason grading of prostate cancer using biopsies: a diagnostic study." The Lancet Oncology 21.2 (2020): 233-241.

[3] Campanella, Gabriele, et al. "Clinical-grade computational pathology using weakly supervised deep learning on whole slide images." Nature medicine 25.8 (2019): 1301-1309. https://www.nature.com/articles/s41591-019-0508-1

3.Deep learning model/architecture or some kind of block /flow diagram to represent the flow of the implementation to be included.

Response: This paper is purely a clinical application paper and it uses the model architecture and training method that was previously elaborated in an earlier paper. To avoid unnecessary repetition, we did not include a block diagram. We do, however, refer the readers to the prior publications that include such diagrams.

Round 2

Reviewer 2 Report

Yes the clarification provided is ok.